# Low RECK Expression Is Part of the Cervical Carcinogenesis Mechanisms

**DOI:** 10.3390/cancers13092217

**Published:** 2021-05-06

**Authors:** Suellen Herbster, Marina Trombetta-Lima, Paulo Thiago de Souza-Santos, Andressa Paladino, Caio Raony Farina Silveira, Mari Cleide Sogayar, Luisa Lina Villa, Ana Paula Lepique, Enrique Boccardo

**Affiliations:** 1Laboratory of Oncovirology, Department of Microbiology, Instituto de Ciências Biomédicas, Universidade de São Paulo, São Paulo 05508-900, Brazil; sueherbster@usp.br (S.H.); andressapaladino@usp.br (A.P.); 2Cell and Molecular Therapy Center (NUCEL), Faculdade de Medicina, Universidade de São Paulo, São Paulo 05508-900, Brazil; m.trombetta.lima@rug.nl (M.T.-L.); mcsoga@iq.usp.br (M.C.S.); 3Leprosy Laboratory, Instituto Oswaldo Cruz, Fiocruz, Rio de Janeiro 21040-360, Brazil; pthiagoss@gmail.com; 4Laboratory of Immunomodulation, Department of Immunology, Instituto de Ciências Biomédicas, Universidade de São Paulo, São Paulo 05508-900, Brazil; caio.silveira@hemocentro.fmrp.usp.br (C.R.F.S.); alepique@icb.usp.br (A.P.L.); 5Department of Biochemistry, Institute of Chemistry, Universidade de São Paulo, São Paulo 05508-900, Brazil; 6Department of Radiology and Oncology, Faculdade de Medicina, Universidade de São Paulo, São Paulo 05508-900, Brazil; l.villa@hc.fm.usp.br; 7Innovation in Cancer Laboratory, Instituto do Câncer do Estado de São Paulo, Universidade de São Paulo, São Paulo 05508-900, Brazil

**Keywords:** cervical cancer, HPV, tumorigenesis, MMP inhibitors, RECK

## Abstract

**Simple Summary:**

Cervical cancer is one of the most lethal types of cancer in women from developing countries. These tumors are caused by long term infection with some human papillomavirus (HPV) types. Commonly, cervical cancer precursor lesions express high levels of matrix metalloproteinases. These enzymes break down specific components of the extracellular matrix affecting tissue structure and stiffness and cell motility. Matrix metalloproteinases and their natural inhibitors, such as Reversion-inducing Cysteine-rich protein with Kazal motifs (RECK) protein, are important in normal tissue maintenance and remodeling and play a major role in the transformation process. Here, we showed that RECK over expression reduced the tumorigenic capacity of cervical cancer cells in vivo. In addition, tumors over expressing RECK presented altered inflammatory infiltrating cells when compared to controls. Our findings are useful to further understand the biology of cervical cancer and can help to determine if RECK may be a good therapeutic target for cervical cancer treatment in the future.

**Abstract:**

Human papillomavirus (HPV)-induced carcinogenesis comprises alterations in the expression and activity of matrix metalloproteinases (MMP) and their regulators. Reversion-inducing Cysteine-rich protein with Kazal motifs (RECK) inhibits the activation of specific metalloproteinases and its expression is frequently lost in human cancers. Here we analyzed the role of RECK in cervical carcinogenesis. Cervical cancer derived cell lines over expressing RECK were used to determine tumor kinetics as well as, cellular, immune and molecular properties in vivo. Besides, we analyzed RECK expression in cervical cancer samples. RECK over expression (RECK+) delayed tumor growth and increased overall survival in vivo. RECK+ tumors displayed an increase in lymphoid-like inflammatory infiltrating cells, reduced number and viability of tumor and endothelial cells and lower collagenase activity. RECK+ tumors exhibited an enrichment of cell adhesion processes both in the mouse model and cervical cancer clinical samples. Finally, we found that lower RECK mRNA levels were associated with cervical lesions progression and worse response to chemotherapy in cervical cancer patients. Altogether, we show that increased RECK expression reduced the tumorigenic potential of HPV-transformed cells both in vitro and in vivo, and that RECK down regulation is a consistent and clinically relevant event in the natural history of cervical cancer.

## 1. Introduction

Worldwide, cervical cancer is the third most frequent and one of the ten most lethal neoplasias among women, accounting for over 565,000 new cases every year. Over 70% of these cases and more than 75% of associated deaths occur in developing countries [1,2]. Cervical cancer is etiologically associated with persistent infection with high-oncogenic risk Human Papillomavirus (HPV) types [3].

HPV-mediated carcinogenesis requires the sustained expression of viral E6 and E7 oncogenes that induce cell immortalization, resistance to apoptosis, evasion of innate and adaptive immune responses and alterations in the expression and activity of extracellular matrix (ECM) components [4,5,6,7]. The direct effect of HPV oncogenes on the expression of different ECM components has been previously reported. A recent study demonstrated that c-Jun inhibition in HPV-transformed cell lines (HeLa and CasKi) was associated with lower metalloproteinases type 9 (MMP-9) mRNA expression levels [8]. Furthermore, Shiau and coworkers showed that HPV16 E6 induced MMP-2 and -9 mRNA expression levels through an IL-8 dependent pathway in H1299 cells [9]. Finally, phosphorylation of HPV18 E7 at S32 and S34 amino acids was associated with the up regulation of MMP-1 and -13 protein expression in C4-1 cervical cancer cells [10]. An extensive description of the main alterations in ECM components during HPV-associated transformation can be found elsewhere [6].

Up regulation of metalloproteinases type 2 and 9 (MMP-2 and MMP-9) expression and activity are the most common ECM related modifications in precursor cervical lesions and invasive carcinoma [6,11,12,13,14]. We reported that this effect correlated with the expression of MMP inhibitors, including Reversion-inducing Cysteine-rich protein with Kazal motifs (RECK) [13,15]. RECK is a serine protease inhibitor that regulates various physiological events including the structural maintenance of ECM, correct formation of blood vessels during embryogenesis, organogenesis, tissue integrity, early neuronal differentiation and limb formation, with RECK deficiency being lethal in mouse embryos [16,17,18]. Besides, reduced RECK expression has been associated with different pathologies, including cancer [19,20,21,22,23,24].

We previously showed that RECK expression is down regulated in most cervical cancer derived cell lines and in cervical lesions [13,25]. Besides, we observed that HPV16 oncogenes expression was associated with reduced RECK protein levels in primary human keratinocytes cultures [15]. Altogether, these observations indicate that RECK inhibition constitutes an important event in cervical carcinogenesis.

In the present study, we show that RECK expression reduces de tumorigenic potential of HPV-transformed cells in vivo and alters the profile of the tumor inflammatory infiltrate. Besides, we show that RECK down regulation is an early trait in cervical cancer history and that RECK levels correlates with protease inhibitors expression, cervical lesions progression, presence of metastases and treatment response in clinical samples.

## 2. Materials and Methods

### 2.1. Cell Culture

Cervical cancer derived cell line SiHa (HPV16 positive; ATCC^®^ HTB-35^TM^, Manassas, VA, USA) and SW756 (HPV18 positive; ATCC^®^ CRL-10302^TM^) were cultured in MEM medium supplemented with 10% fetal bovine serum (M10). Immortalized human embryonic kidney cells (HEK293T; ATCC^®^ CRL-3216^TM^) were cultured in DMEM medium supplemented with 10% fetal bovine serum (D10). All cell culture procedures were performed at 37 °C and 5% CO_2_. For cell proliferation analysis cells were seeded in 24 well plates (1.0 × 10^4^ cells/well) and cultured for 1 to 8 days. Cells were counted every 24 h using a hemocytometer.

### 2.2. Obtention of Cells over Expressing RECK

The lentiviral vector p156RRLsinPPCCMVIns3IRESPRC (pLV EGFP), described elsewhere, was used for RECK over expression [26]. Lentiviral particles were obtained by transfecting packaging plasmids p59, p60 and p61 [27] into HEK293T cells. For each transfection, 1.0 × 10^6^ cells were plated in 100 mm Petri dishes with 10 mL of D10 medium and incubated for 24 h. Then, 2.0 µg of each plasmid p59, p60 and p61 and 2.0 µg of the lentiviral vector were added to 12 µL of Lipofectamine^TM^ 3000 reagent (ThermoFisher, Waltham, MA, USA) and 16 µL of P3000^TM^ reagent supplied by the manufacturer. Transfection efficiency was monitored in a fluorescence microscope after 24 and 48 h (EVOS^®^FL AMG, ThermoFisher Scientific, Waltham, MA, USA) for EGFP signal detection. After 48 and 72 h cultures supernatants containing the lentiviruses were collected and stored at −80 °C until use.

For lentiviral transduction, 2.5 × 10^4^ SiHa or SW756 were seeded in 96-well culture plates with 200 μL of M10 medium for 24 h and transduced with 200 μL of lentivirus stock and polybrene (4 μg/mL) (Sigma-Aldrich, Saint Louis, MO, USA) for 24 h. Transduction efficiency was monitored after 24 and 48 h by EGFP signal detection in a fluorescence microscope (EVOS^®^FL, AMG, USA). Finally, aseptic fluorescence-activated cell sorting (FACS) was performed in a cytometer (BD FACSaria^TM^ II U, San Jose, CA, USA) to isolate EGFP+ SiHa or SW756 transduced cells (Appendix A).

### 2.3. Western Blot

Total protein extracts from monolayer cell cultures were prepared by lysis in RIPA buffer (150 mM NaCl, 50 mM Tris HCl-pH 7.5, 0.5% NP-40, 0.1 mM EDTA) supplemented with protease inhibitors (Complete Mini, Roche Diagnostics).

Altogether, 60 µg of total protein extracts were subjected to 10% SDS-PAGE polyacrylamide gel electrophoresis using the mini-Protean II Cell system (Bio-Rad, Hercules, CA, USA) and transferred to a polyvinylidene difluoride membrane (PVDF, Hybond-P, Cytiva, MA, USA). Protein expression was detected using anti-RECK protein (110 kDa, Cell Signaling CS3334, 1:1000 dilution), and anti-β-Tubulin (55 kDa, ThermoFisher, Waltham, MA, USA, 1:10,000 dilution) antibodies, horseradish peroxidase (HRP)-conjugated secondary antibodies and revealed using Enhanced Chemiluminescence procedures (Cytiva, Marlborough, MA, USA).

### 2.4. Clonogenic, Migration, Invasion and Anchorage Independent Growth Assays

Cells were seeded in six-well plates (1.0 × 10^2^ cells/well) and cultured in M10 medium for 15 days. For migration and invasion assays 5 × 10^4^ cells/chamber suspended in MEM were added to the upper compartments of the Transwells with 8 μm pores (Corning Incorporated, Corning, NY, USA). M10 was added to the wells for chemotactic attraction, followed by incubation of the plates for 12 h for migration and 36 h for invasion assay. Cell invasion was assessed by their ability to transpose both a Matrigel^®^ coat (BD Biosciences, San Jose, CA, USA) and barrier (Corning Incorporated, USA). Colonies, migrating and invading cells were fixed with 70% ethanol solution (Merck Millipore, Darmstadt, Germany), stained with 0.5% crystal violet (ThermoFisher, Waltham, MA, USA) in 10% ethanol solution and counted. For anchorage independent growth assay, 2.5 × 10^2^ cells were suspended in 500 μL of M10 medium with 0.6% agarose and seeded in 24-well plates coated with the 1% agarose, and covered with M10 medium. After 30 days 50 μL of 3-(4,5-dimethylthiazol-2-yl)-2,5-diphenyltetrazolium bromide (MTT, 5mg/mL) was added for staining and counting of colonies.

### 2.5. Tumor Spheroids Formation

Spheroid formation was assessed with culture of 5 × 10^3^ (SiHa or SW756) seeded over 50 µL 1% agarose in PBS coat in U bottom 96 well plates. Spheroids were evaluated after seven days. Proliferation and cell viability were evaluated, after mechanical disruption, by cell count with Trypan blue 0.4% (ThermoFisher, Waltham, MA, USA) dye 1:10, using hemocytometer.

### 2.6. In Vitro Zymography

Monolayer cell cultures were maintained in FBS free MEM for 72 h, when the supernatants were harvested and centrifuged at 1000× *g* for 10 min at 4 °C. Supernatants were stored at −80 °C until use. Zymography was performed according to DQ™ Collagen, type IV from Human Placenta, Fluorescein Conjugate protocol (Thermo Fisher Scientific, MA, USA) and as previously described [28]. Briefly, 100 µL of FBS free supernatant was mixed with 100 µL of reaction solution (50 nM Tris, 5 mM CaCl_2_-pH 7.5, 1 µM ZnCl_2_ and 50 µg/mL DQ™ Collagen kept in 0.03% sodium azide) in black 96 wells plates. After two hours in the dark at 37 °C the fluorescence was measured in Glowmax fluorimeter with a 490/510 nm filter.

### 2.7. SDS-PAGE Gelatin Zymography

Cell culture supernatants (30 µg) were subjected to non-denaturing electrophoresis in 5% gelatin SDS-PAGE in 65 V for 30 min followed by 90 V for two hours at room temperature. After, the gel was washed twice with 2.5% Triton X-100 solution for 15 min at room temperature followed by 48 h of incubation with the reaction solution (50 nM Tris, 5 mM CaCl_2_-pH 7.5, 1 µM ZnCl_2_) at 37 °C. Then, the gel was stained using Coomasie brilliant blue R-250 0.5% for 5 min followed by sequential incubations with Methanol/Acetic acid (10%:10%) solution until bands were observed.

### 2.8. Tumor Growth Kinetics in Nude Mice

All experiments involving mice were carried out under SPF conditions in the Isogenic Mice Animal Facility in the Department of Immunology at University of São Paulo. This project was approved by the Ethics Committee on the Use of Animals (CEUA, protocol number 138, page 13 from book 03) of the ICB/USP. For tumor analysis 2 × 10^6^ cells were injected *s.c.* on the dorsal flanks of female nude mice (10 animals per group). Tumor diameter was measured once a week with manual caliper and tumor volume was calculated using the formula V = (D × d^2^)/2, where V is tumor volume, D is the largest diameter measured and d the smallest diameter measured. Tumors were allowed to grow up to 500 mm^3^, when the animals were euthanized. Tumors were collected immediately and processed for the different types of analysis.

### 2.9. Analysis of Intratumoral Cell Populations

Single cell tumor suspensions were prepared by mincing the tumors with a scalpel and digestion of the fragments in 1 mg/mL Collagenase I and IV diluted in Hank’s solution (15 mmol/L HEPES-pH 7.4, 0.5 U/mL DNase (Worthington, Columbus, OH, USA) and 5% fetal bovine serum) at 37 °C and 1300 rpm agitation. Cells were then filtered through a 70 μm strainer. Single cells (1 × 10^6^) were suspended and labeled for surface antigens with specific antibodies (Antibodies references in Appendix A), followed by cell fixation, permeabilization, DAPI staining and analysis by flow cytometry. Cell cycle analysis was performed using DAPI staining (10 μg/mL) as previously described [29]. Suspended labeled cells were fixed in 1% fresh buffered formaldehyde solution overnight, and labeled with 10 μg/mL DAPI for 30 min in the dark. Flow cytometry data acquisition was performed in the BD FACSCanto™ II with Diva software (BD Biosciences, San Jose, CA, USA). Data analysis was performed using FlowJo software (BD Biosciences, San Jose, CA, USA).

### 2.10. In Situ Zymography

SW756 tumors were fresh frozen in TissueTek (Sakura Finetek, Torrance, CA, USA) and stored at −20 °C. Twenty µm thick tumor cryosections were covered with 200 µL of the reaction solution described before and kept in the dark at 37 °C for 4 h. After washing with PBS1X saline solution, DAPI counterstaining [29] was performed and the slides were mounted for immediate analysis with IX70 Olympus fluorescence microscope (Olympus, Corp., Tokyo, Japan) and images were obtained with a DP70 Olympus camera, using its own software.

### 2.11. Immunofluorescence

A total of 10 µm thick frozen tumors slices were immunostained with anti-RECK monoclonal antibody (1:1000) for two hours at 37 °C. After washing with PBS, the slides were incubated in the presence of an anti-rabbit (AlexaFluor 594-111-585-003, Jackson ImmunoResearch, West Grove, PA, USA) secondary antibody for two hours in the dark at 37 °C and counterstained with DAPI.

### 2.12. Protein Arrays

Total protein extracts of SW756 control and RECK+ tumors cells were obtained using specific extraction buffer for each Proteome Profiler^TM^ Array (ARY004B; ARY007; ARY012; ARY015 and ARY025, R&D Systems, Minneapolis, MN, USA) according to manufacturer’s instructions. Protein concentration was determined using Pierce^TM^ BCA Protein Assay Kit (Thermo Fisher Scientific, Waltham, MA, USA).

### 2.13. Global Gene Expression Analysis

Clinical data from cervical samples named GSE7803, GSE9750, GSE51993, GSE75132, GSE39001, GSE26511, GSE3578 were retrieved from the Gene Expression Omnibus online platform (GEO http://www.ncbi.nlm.nih.gov/geo, accessed on 14 April 2021). All data sets were analyzed for possible outliers using the site https://graphpad.com/quickcalcs/grubbs1/, accessed on 14 April 2021.

### 2.14. GO Pathway Enrichment Analysis

Enrichment analysis was carried out with R package GOstats [30]. Analyses were followed by application of hypergeometric test and only biological processes with *p* < 0.001 were considered to be enriched.

### 2.15. Statistical Analysis

Data were analyzed using Mann–Whitney, Kruskal–Wallis or two-way ANOVA for nonparametric tests. All statistical analyzes were performed with GraphPad Prism version 5.00 for Windows (GraphPad Software, La Jolla, CA, USA). A *p*-value of less than 0.05 was considered statistically significant.

## 3. Results

### 3.1. RECK Expression Delays HPV16- and HPV18-Positive Tumors Growth and Increases Overall Survival In Vivo

Previous results from our laboratory indicate that RECK down regulation is important in cervical cancer natural history. Here, we performed a series of experiments in order to determine the impact of RECK expression in tumor growth kinetics in vivo. First, we injected RECK+ cervical cancer derived cells in nude mice and interrupted the experiment when tumors from the control group reached approximately 500 mm^3^. At this point, animals injected with RECK+ cells exhibited tumors with reduced volume, when compared to controls (Figure 1A). In parallel, we executed an independent experiment where we followed tumor growth for up to 150 days. Whenever the tumor volume reached 500 mm^3^, the animal was removed/censored from the survival analysis, according to humane treatment guidelines. We observed that tumors established from SiHa RECK+ or SW756 RECK+ cells exhibited delayed tumor detection compared to those obtained from control cells (Figure 1B,C). Besides, animals injected with SiHa RECK+ and SW756 RECK+ displayed delayed tumor growth kinetics (Figure 1D,E) and, consequently, higher survival (Figure 1F,G) compared to controls. Importantly, only 20% of SiHa RECK+ injected animals presented tumors of approximately 500 mm^3^ after 150 days of follow up (Figure 1F). Importantly, 30% of the animals injected with SW765 RECK+ cells exhibited tumor regression (Figure 1C). Our results show that high levels of RECK delayed the establishment and growth of tumors derived from HPV16- and HPV18-transformed cell lines while increasing animal survival.

### 3.2. RECK Expression Reduces Characteristics Associated with the Tumorigenic Potential of HPV-Transformed Cell Lines

The data described above clearly show the existence of differences in the growth dynamics between tumors derived from control and RECK+ cells. There are, at least, two not mutually exclusive possibilities to explain these results: (a) differences in the tumorigenic potential between control and RECK+ cells and, (b) differences in the tumor microenvironment of tumors established from control and RECK+ cells. To address these possibilities, we first performed in vitro experiments to characterize the effect of RECK expression on HPV-transformed cells SiHa and SW756 (Appendix A). We observed that, neither the cell doubling time (CDT) nor the clonogenic potential of these cells were affected by the ectopic expression of RECK (Appendix A). SiHa RECK+ cells exhibited a mean 59.2 ± 13.3% reduction in chamber migration when compared to control cells (Figure 2A), while no effect was observed for SW756 cells. Conversely, we observed a consistent reduction in chamber invasion for SiHa RECK+ (51.2 ± 24.9%, *p* = 0.0043) and SW756 RECK+ (58.3 ± 17.8%, *p* = 0.0080) when compared to their specific controls (Figure 2B). Furthermore, RECK+ cells exhibited a significant reduction in anchorage independent growth capacity (Figure 2C). Besides, we observed a marked drop in the number of cells per spheroid obtained from RECK+ cells (Figure 2D). In addition, SW756 RECK+ spheroids displayed higher percentage of non-viable cells when compared to controls (Figure 2E). Finally, we detected a significantly lower proteolytic activity in the supernatant of both SiHa RECK+ and SW756 RECK+, when compared to their matched control cells (Figure 2F and Appendix A). Altogether, these results suggest that over expression of RECK negatively affects invasion, substrate independent growth, spheroid formation potential and collagen IV degradation activity in the HPV-transformed cell lines tested in this study.

### 3.3. Tumors Established from RECK+ Cells Exhibited Less Tumor and Endothelial Cells and an Increased Inflammatory Infiltrate Population

Besides a direct effect of RECK expression on cell oncogenic potential as shown before (Figure 2), we hypothesized that over expression of this protein may induce changes in the microenvironment that may affect tumor establishment and progression in vivo. In order to address this issue, we analyzed the different cell populations present in the tumor microenvironment (TME) of control and RECK+ tumors. Tumor cell populations were labeled with antibodies and analyzed by flow cytometry to identify leukocytes (CD45+/CD31−), endothelial cells (CD45−/CD31+) and tumor cells (CD45−/CD31−) (Appendix A). We observed that tumor cells were the most abundant cell population in both SiHa and SW756 tumors (93.6 ± 0.6% and 68.2 ± 4.6%, respectively). On the other hand, SiHa RECK+ and SW756 RECK+ tumors exhibited a lower proportion of tumor cells (49.7 ± 11.5% and 37.6 ± 12.7%, respectively) (Figure 3A). Besides, SW756 RECK+ tumor cells presented reduced viability when compared to control tumors, as indicated by detection of an increase in the sub-G1 population (Figure 3B). This effect was not observed in SiHa RECK+ tumors (Figure 3B). Importantly, both SiHa RECK+ (1.8 ± 0.2% vs. 3.9 ± 0.5% in control) and SW756 RECK+ (5.5 ± 1.7% vs. 22.3 ± 6.4% in control) tumors exhibited a reduced proportion of endothelial cells, with reduced viability (Figure 3A,C).

Interestingly, both SiHa RECK+ (48.5 ± 11.6% vs. 2.5 ± 1.1% in controls) and SW756 RECK+ (56.9 ± 14.1% vs. 9.5 ± 2.1% in controls) tumors showed striking increase in the proportion of inflammatory cells (CD45+) infiltrating the tumors when compared to their controls (Figure 3A). The absolute number of inflammatory cells per 100,000 tumor cells was also increased in RECK+ tumors (Figure 3D). Finally, SiHa RECK+ tumors also showed increased inflammatory infiltrate viability (Figure 3E).

### 3.4. RECK Expression Induces Alterations in the Intra-Tumoral Inflammatory Infiltrate Profile

We then assessed the frequency of macrophages (Ly6C+/F4/80+), neutrophils (Ly6G+), double negative cells (Ly6C−/Ly6G−), NK cells (Ly6C−/Ly6G−/CD49b+) and potential myeloid derived suppressor cells (MDSC) (Ly6C+/Ly6G+), within the CD45+ leukocyte population infiltrating the tumors (Appendix A). Although the major myeloid population infiltrating SiHa tumors were macrophages and SW756 tumors were MDSC, we believe their role may be redundant in the tumor microenvironment [31]. Of notice, however, was that RECK+ tumors exhibited a significant reduction in the frequency of neutrophils and of potential MDSC populations (Figure 4A).

Moreover, we found that RECK+ tumors presented an increased number of double negative population and NK cells per 100,000 tumor cells than control tumors (Figure 4B,C). Histological analysis of tumor sections confirmed the presence of cells with monocyte/macrophage morphology in control tumors (Figure 4D) and detection of an increased number of cells with lymphocytic morphology in RECK+ tumors (Figure 4D). This result was further confirmed with the significant increase in low FSC/low SSC population in RECK+ tumors compared to controls (Appendix A). Altogether, these results suggested that RECK over expressing tumors presented a marked enrichment of a lymphoid-like inflammatory infiltrate population which included NK cells, whilst showed a significant decrease in potential MDSC population when compared to control tumors.

### 3.5. RECK Expression Alters Collagenase Activity, Collagen Content and Expression of Protease Inhibitors In Vivo

We next assessed in situ collagenase activity to determine whether RECK+ was associated with alterations in the ECM (Appendix A). These analyses were performed using SW756 RECK+ tumors, since SiHa RECK+ tumors were too small to retrieve enough material for analysis. We detected lower collagenase activity in RECK+ tumors when compared to controls (Figure 5A,B and Appendix A). Finally, we observed that SW756 RECK+ tumors exhibited an evident increase of collagen fibers when compared to controls as determined by Picro Sirius staining (Figure 5C,D and Appendix A).

To further characterize the effects of RECK over expression in our model, we analyzed the expression of 353 proteins associated with modulation of both human and murine angiogenesis, human proteases and their inhibitors, human phospho-immunoreceptors and non-hematopoietic receptors. We detected 52 differentially expressed proteins in RECK+ tumors compared to controls (Figure 5E, Appendix A). Interestingly, we found increased levels of several protease inhibitors in RECK+ tumors, including members from the Serpin and Cystatin family and the Tissue Inhibitor of Metalloproteinases 2 (TIMP2). This data was analyzed for Kyoto Encyclopedia of Genes and Genomes (KEGG) pathway enrichment using DAVID web tool. We observed that 8 out of 52 differentially expressed proteins in RECK+ tumors presented functions associated with cell adhesion modulation (Figure 5E, bottom). Finally, several cellular receptors were differentially expressed in RECK+ tumors, including ALCAM/CD166, CD40/TNFRSF5, CXCL8/IL-8, NOTCH-1 and PAR-1.

### 3.6. RECK Expression Correlates with Higher Levels of Protease Inhibitors Both in Experimental Mouse Models and Cervical Cancer Clinical Samples

We then sought to determine if the correlations between RECK expression and other protease inhibitors, described above, could extend to the mRNA expression in cervical cancer samples. We found a moderate but significant positive correlation between RECK mRNA and SerpinE1 and TIMP2 mRNA levels in cervical cancer samples (Figure 6A,B). Besides, we observed that TIMP2 expression also presented a moderate to strong positive association with RECK levels in both HPV-positive and HPV-negative HNSCC clinical samples, while SerpinE1 mRNA levels did not correlate with RECK expression in these samples (Figure 6A,B). We then used the expression data retrieved from the TCGA platform to investigate GO pathway enrichment. None of the enriched biological processes found were exclusive to HPV-associated cancers. However, we observed that RECK associated enrichment for collagen fibril organization, extracellular matrix organization, tube morphogenesis and response to growth factor pathways were common to all three groups analyzed in this study (Figure 6C,D, red arrows; and Appendix A). Similar to KEGG pathway enrichment data from our mouse models, we showed here that the cell adhesion process was associated with RECK mRNA expression in cervical cancer (Figure 6C, orange arrow).

### 3.7. RECK down Regulation Is a Consistent Trait in Cervical Cancer History

Previous data from our laboratory and the results presented in this study clearly indicate that RECK plays a role in cervical carcinogenesis. Therefore, we decided to confirm our observations in a higher number of clinical samples. We first determined the levels of RECK mRNA in a series of cervical cancers or cervical intraepithelial neoplasias (CIN) datasets available in GEO open-access database. High-grade CIN (CIN2/3) are alterations in the cervical epithelium and are considered the precursor lesions of cervical cancer. We observed that RECK gene expression was consistently down regulated in CIN, when compared to normal tissue in both GSE51993 (Figure 7A) and GSE7803 (Figure 7B) datasets. Similarly, we observed reduced RECK expression in invasive cervical carcinomas (CC), when compared to normal cervical tissue in both GSE7803 (Figure 7B) and GSE39001 (Figure 7C) datasets.

We next analyzed RECK gene expression according to cervical disease outcomes. We observed lower levels of RECK expression in HPV16 positive CIN lesions which progressed to CIN3 or invasive cancer, when compared to cervical tissue from HPV negative healthy subjects (GSE75132) (Figure 7D). However, no differences in RECK levels were observed between HPV negative women and women with CIN which did not progress to CIN3+ (Figure 7D). Moreover, we found that RECK expression was down regulated in early-stage CC patients who presented pelvic lymph node metastasis when compared to controls (GSE26511) (Figure 7E). Finally, CC patients who showed better response to neoadjuvant chemotherapy (NAC) followed by radical hysterectomy (RH) expressed higher levels of RECK (GSE70035) (Figure 7F) [32].

## 4. Discussion

RECK protein is involved in the suppression of tumor invasion, angiogenesis and metastasis. In part, these effects are associated with RECK’s function as a specific inhibitor of MMP-2, MMP-9, MT1-MMP and extracellular MMPs ADAM10 and CD13/Aminopeptidase [24,33]. Previous observations indicate that loss or reduction of RECK expression is a common trait in several types of human tumors. This event is often associated with worse prognosis and increased metastasis in several human cancers [16,21,26,34,35,36,37,38,39,40,41,42,43,44,45]. We previously reported the existence of an inverse correlation between RECK expression and cervical disease progression [15]. Here, we confirmed that RECK down regulation is an early and consistent event in the natural history of cervical carcinogenesis using expression data from different studies carried out by diverse groups in different centers. We also found that higher RECK expression correlated with better response to chemotherapy in CC patients.

The results presented here clearly support the existence of a negative effect of RECK over expression on the tumorigenic potential of HPV-transformed cells. These results are in line with those reported in a recent study that showed that RECK over expression in cervical cancer derived cell lines (C33A and HeLa) was associated with reduced migration and invasion potential [46]. Reduced volume of RECK+ tumors may be consequence of a drop in cell viability and TME alterations as observed for SW756 RECK+ tumors. Interestingly, SW756 RECK+ cells showed decreased cell viability in both 3D cell culture and in vivo models.

The local impact of low RECK expression is primarily due to increased proteolytic activity of specific MMPs, MMP-2, -9 and -14. In the context of tumor neoangiogenesis, the higher activity of MMP-2 and -9 was consistently associated with the release of VEGF from ECM reservoirs, contributing to the angiogenic switch [47,48,49,50]. Low RECK expression was previously correlated with increased capacity of vessel formation, invasion and onset of local and distant metastases in animal models of breast tumor [44]. Here, we showed that RECK+ tumors presented reduced tumor and endothelial cell populations and increased inflammatory infiltrate.

Strikingly, we observed that RECK over expression was associated with a robust leukocyte recruitment to the TME and also differences in the cell lineages composing the infiltrate. This is the first evidence, as far as we know, that RECK expression can interfere with the inflammatory infiltrate associated to the TME. In addition, we found that inflammatory cells infiltrating SW756 RECK+ tumors presented reduced cell size and lymphoid morphology, as opposed to controls. The cell population detected at higher frequency in RECK+ tumors could be innate lymphoid cells (ILC) or B lymphocytes, both of which are present in nude mice. It is unlikely that B lymphocytes would display antitumor activity. However, depending on the phenotype of the ILC population, it would be possible for these cells to display anti-tumor activity [51].

Previous reports showed that MMP and Galectin-3 expression and function may interfere with tumor associated NK cell cytotoxicity. In fact, MMP-9 associated cleavage and consequent shedding of tumor cell membrane proteins such as MIC-A, MIC-B and ULBP-2, lead to reduced NK associated tumor cell death in gastric cancer, lung adenocarcinoma and osteosarcoma [52,53,54]. Therefore, it is tempting to speculate that RECK expression could preserve NK cells as observed in our model.

Tumor associated neutrophils (TAN), macrophages (TAM) and myeloid derived suppressor cells recruitment can be mediated by IL-8 in different malignancies including cervical cancer [55,56,57,58,59,60,61,62,63,64]. Others have previously described that high levels of IL-8 were associated with HPV persistence, cervical lesions progression, increased tumor cell proliferation and worse prognosis [64,65,66,67,68]. Moreover, induction of IL-8 secretion was associated with higher levels of MMP-2 and MMP-9 activity in endothelial cell and bladder cancer models [69,70,71]. Walsh and collaborators reported that RECK+ is associated with a threefold decrease in IL-8 mRNA levels and that RECK can physically bind to IL-8 in a breast cancer model [44]. Here, we report that RECK over expression is associated with reduced intra-tumoral levels of IL-8 in SW756 tumors. Moreover, MMP-9 associated IL-8 cleavage leads to a tenfold increase in neutrophil activation and recruitment to the infection site [72]. It remains to be determined whether the RECK–IL-8 axis reported in our study contributes to the decreased TAN, TAM and potential MDSC populations observed in SW756 RECK+ tumors.

TIMP2 over expression has also been associate with reduced MDSC recruitment, tumor volume and angiogenesis in a lung adenocarcinoma model [73]. Here, we show for the first time, that RECK over expression is associated with higher levels of the TIMP2 protein in SW756 tumors, therefore we argue that RECK–TIMP2 cooperate to reduce the recruitment of potential MDSC in our model.

Alternatively, others have shown the role of RECK in mobilization/retention of HSPC (hematopoietic stem and progenitor cells) in the bone marrow. It has been observed that RECK associated MT1-MMP inhibition is crucial in reducing the migration potential of HSPC cells out of the bone marrow [74,75]. The potential interference of RECK with the antitumor response warrants a deeper exploration in immunocompetent animal models. This approach will be crucial to determine the potential role of RECK as an immunotherapeutic target for the treatment of cervical cancer.

We also described that RECK over expression is associated with differential expression of several proteins involved with MMP activity and cell adhesion in SW756 derived tumors. The expression of SerpinE1/PAI-1 and TIMP2 targets also correlated with RECK mRNA levels in cervical cancer samples. Interestingly, these proteins also present specific MMP inhibition roles which may cooperate with RECK function to regulate MMP activity.

The biological processes associated with RECK expression, accessed with GO, are aligned with RECK known functions in MMP activity inhibition, proper vessel formation modulation and interference with growth factor response, such as EGFR pathway [39,76,77]. Cell adhesion regulation was one of most enriched pathways in RECK+ tumors in mice and, also, in cervical cancer patients. We argue that RECK over expression leads to improved tumor cell adhesion, which may contribute, in part, to the observed reduced tumor volume in mice.

## 5. Conclusions

The data presented here clearly showed that down regulation of RECK is an important early step in CC natural history and that RECK activation may be a promising therapeutic target for CC. The current study has not yet addressed the mechanisms by which RECK expression is reduced along the development of cervical cancer. Data from our laboratory indicates that this effect may depend, at least in part, on a direct effect of HPV early proteins on the RECK promoter (manuscript in preparation).

Finally, we generated a model for the RECK+ effect on cervical cancer. RECK+ was associated with (i) delayed tumor establishment, (ii) reduced tumor growth and volume, (iii) reduced endothelial and tumor cell populations, (iv) increased proportion of specific inflammatory cells, (v) higher collagen content and (vi) increased levels of several protease inhibitors, including TIMP2 and SerpinE1/PAI-1. RECK over expression also correlated with higher protein levels of CD40/TNFRSF5 and lower levels of IL-8 protein in a SW756 derived tumor mice model (Figure 8). Taken together, the data presented here may contribute to further understand the role of RECK in HPV-mediated cervical carcinogenesis.

## Figures and Tables

**Figure 1 cancers-13-02217-f001:**
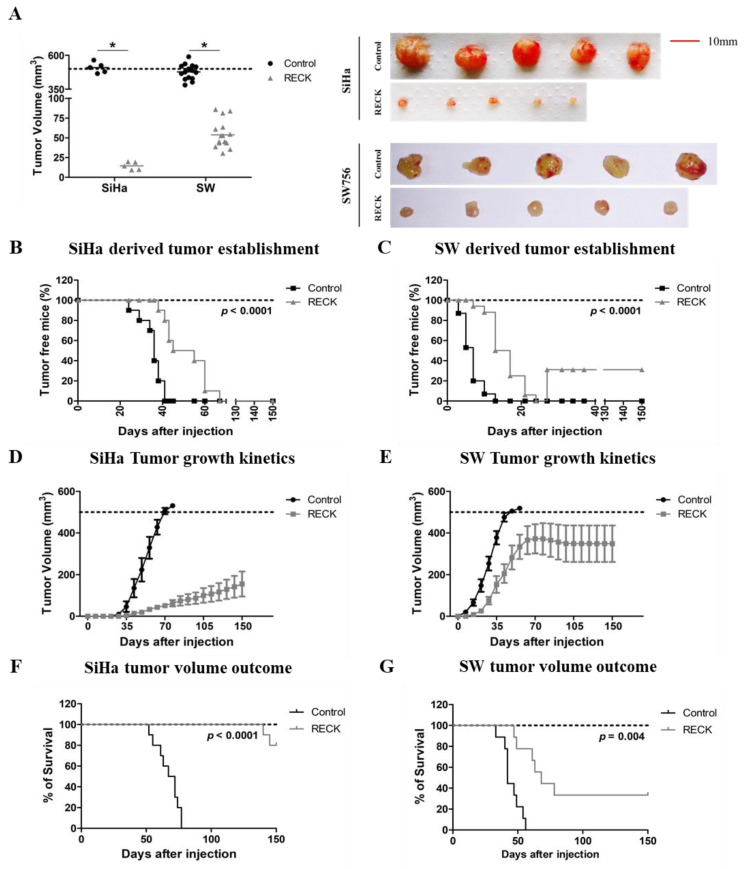
RECK over expression is associated with delayed SiHa and SW756 derived tumor establishment, reduced tumor growth and increased survival in nude mice. (**A**). Differential tumor growth experiment in nude mice. Here, the experiment ended once most of the animals from the control group presented tumors with 500 mm^3^. (**B**–**G**). The effect of RECK over expression on cervical cancer derived cells tumor establishment and overall survival was assessed in vivo. SiHa (**B**,**D**,**F**) and SW756 (**C**,**E**,**G**) cells over expressing RECK were subcutaneously injected in nude mice (2 × 10^6^ cells/mouse). Tumors were identified as established once they were measurable with a manual caliper and the kinetics of tumor establishment for each of the cell lines tested are illustrated in B and C. D and E. Representative data from a tumor growth kinetics experiment. Tumor volume was measured weekly for at least 150 days. Overall survival was determined as each animal presented a tumor volume of 500 mm^3^ during the 150 days of experiment (F and G). * *p* < 0.05 by Mann–Whitney (1A), Log rank (1B, 1C, 1F and 1G) or two-way ANOVA (1D and 1E) test.

**Figure 2 cancers-13-02217-f002:**
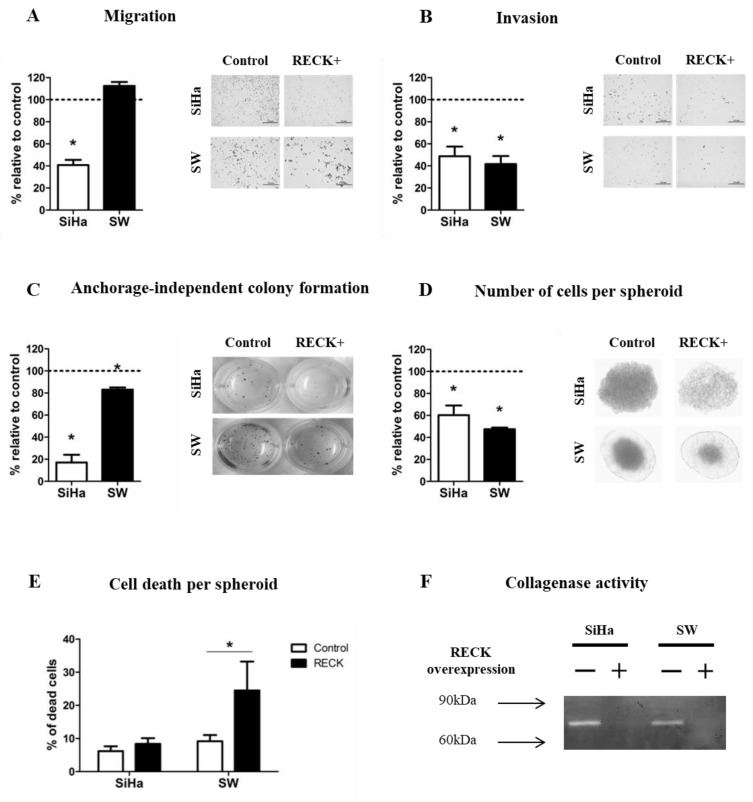
RECK over expression reduces SiHa and SW756 cell lines invasion potential and 3D growth. (**A**,**B**) Chamber migration and invasion assays. Cells present in 10 independent fields per chamber were counted. (**C**) Anchorage independent growth assay. After 30 days in culture, three-dimensional colonies were stained using MTT and all visible colonies were counted. (**D**,**E**) Spheroid formation assay. After 7 days in culture, spheroids were dissociated, stained with Trypan blue and counted. (**F**) Gelatinase activity assay. Zymography was carried out by subjecting 30 ug of protein to SDS-PAGE supplemented with 5% gelatin. After electrophoresis, the gel was incubated in an activation solution at 37 °C for 48 h, the gel was then stained with 0.5% crystal violet and images were developed with use of MultiDoc-It^TM^ 120 Imaging System (Analytic Jena, Upland, CA, USA). All experiments were carried out in triplicates. * *p* < 0.05 by Mann–Whitney (2A–E) test.

**Figure 3 cancers-13-02217-f003:**
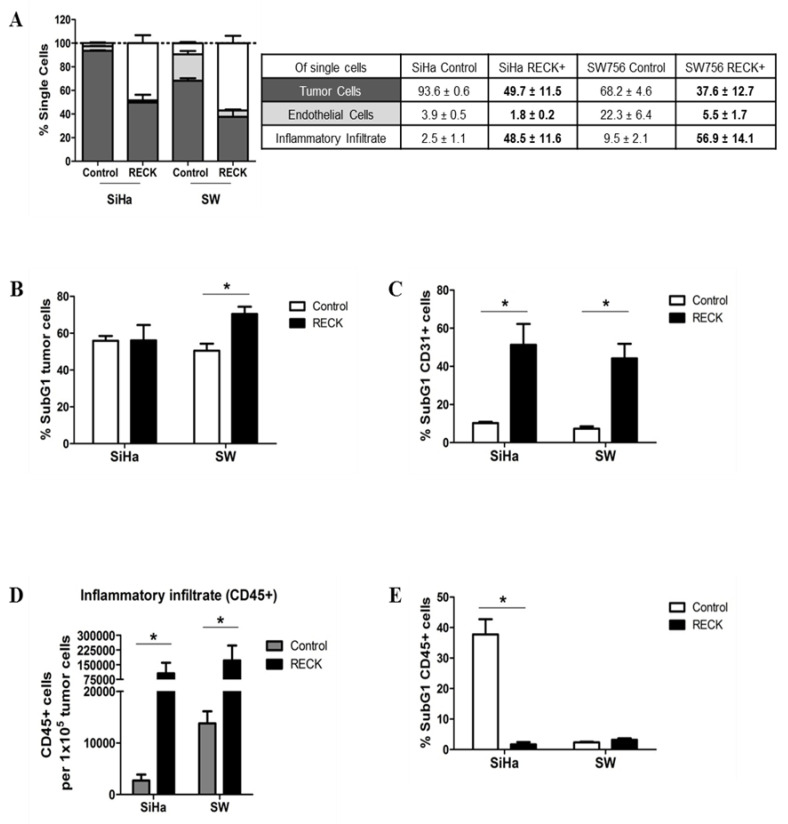
RECK over expression is associated with alterations in tumor microenvironment content. The injection protocol for tumor establishment was the same as described in Figure 1A. Here, the experiment ended once most of the animals from the control group presented tumors with 500 mm^3^. (**A**). Intra-tumoral populations were characterized by immunostaining with anti-CD31 and CD45 antibodies, followed by flow cytometry. Tumor fractions of at least three animals from each group were used for the analysis and at least 1 × 10^5^ cells/events were acquired. The EGFP+ population was identified as tumor cells. CD31+ and CD45+ populations identified endothelial cells and inflammatory infiltrate, respectively. Significant differences in population proportion are indicated in bold font in the table. (**B**,**C**,**E**) Tumor (**B**), endothelial (**C**) and inflammatory (**E**) cell populations stained with DAPI were evaluated on hypodiploid DNA content (SubG1) using flow cytometry. (**D**) Local inflammatory infiltrate population relative to 100,000 tumor cells. * *p* < 0.05 by Kruskal–Wallis (3A) or Mann–Whitney (3B–E) test.

**Figure 4 cancers-13-02217-f004:**
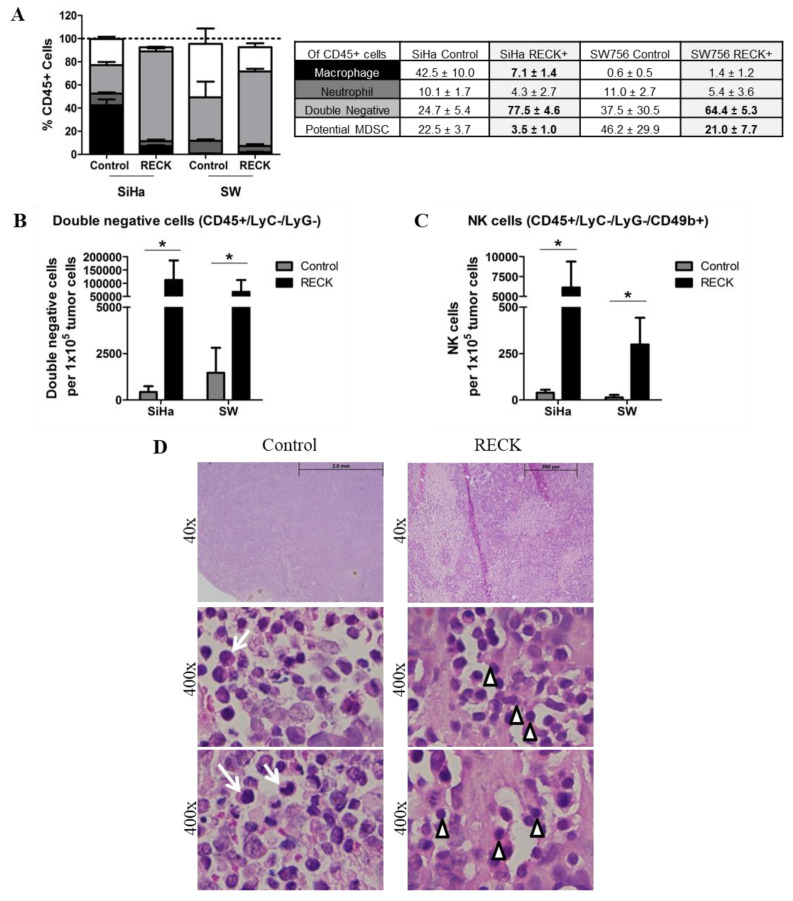
RECK over expression is associated with increased proportion of specific inflammatory infiltrate cell populations. (**A**). Intra-tumoral populations were characterized by immunostaining with CD45, Ly6C, Ly6G, F4/80 and CD49b antibodies, followed by flow cytometry. Tumor fractions of at least three animals from each group were used for the analysis and at least 1 × 10^5^ cells/events were acquired. Significant differences in population proportion are indicated in bold font in the table. (**B**,**C**). Proportion of Double negative (**B**) and NK cells (**C**) relative to 100,000 tumor cells. (**D**). Representation of hematoxylin/eosin tumor sections used for morphology evaluation of the inflammatory infiltrate in SW756 RECK+ tumors. White arrows identify cells with monocyte and macrophage morphology and white arrow heads indicate lymphoid phenotype cells. * *p* < 0.05 by Kruskal–Wallis (4A) or Mann–Whitney (4B and 4C) test.

**Figure 5 cancers-13-02217-f005:**
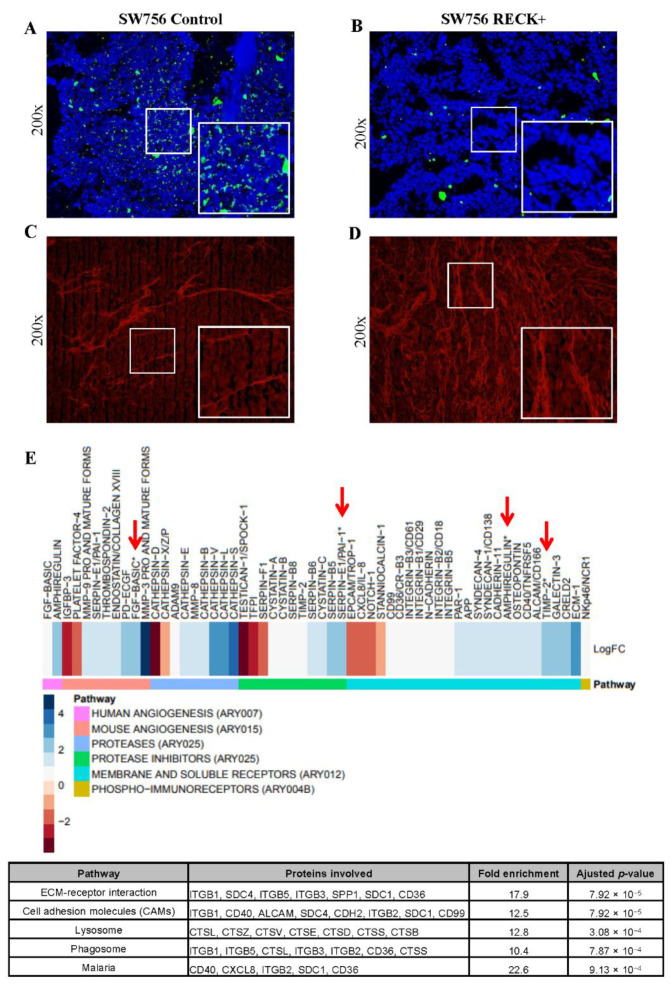
SW756 RECK+ tumors present collagenase activity, higher collagen content and increased levels of collagenase inhibitors. (**A**,**B**). In situ zymography assay carried out with SW756 tumors using DQ Collagen IV (green signal represents collagen breakdown). (**C**,**D**) Representative images showing Picro Sirius fluorescent staining of SW756 tumors (red signal shows collagen fibers). All fluorescent assays were carried out in the presence of DAPI staining and the merged images were processed using Image J software. (**E**). Heatmap representation of differentially expressed proteins in RECK+ tumors when compared to control. The fold change is expressed in log_2_ scale; where blue and red scale represents increased and decreased relative expression, respectively. Antibody arrays were depicted in different colors on the far-left side, below the heatmap. Proteins were identified as differentially expressed when fold change, expressed in log_2_, was over 0.58 or under −0.58. Red arrows show differentially expressed proteins detected in more than one antibody array. The table displays the enrichment analysis in KEGG pathway database with use of DAVID web tool. Statistical analysis was performed with Fisher’s exact test (bottom).

**Figure 6 cancers-13-02217-f006:**
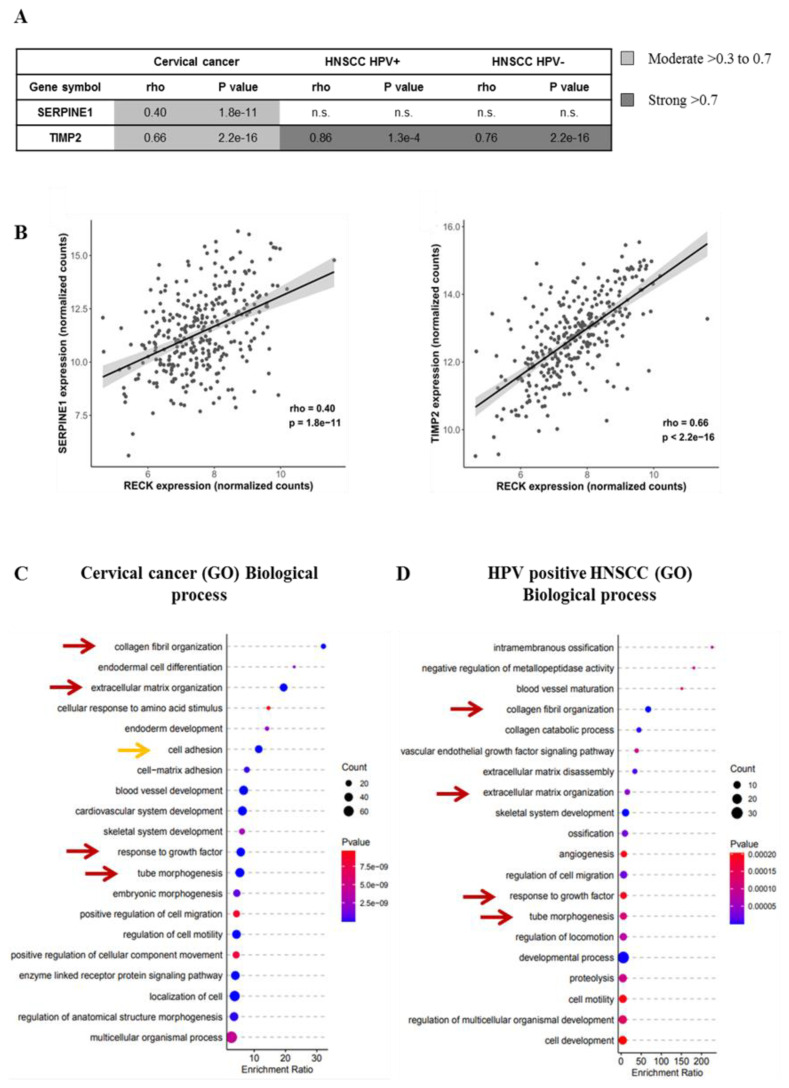
Extracellular matrix remodeling is one of the most enriched biological processes associated with RECK expression in both cervical cancer and HNSCC patients. (**A**). Correlation between SerpinE1 or TIMP2 mRNA levels and RECK mRNA expression in cervical cancer and HNSCC clinical samples deposited in TCGA platform. (**B**). Representative image of SerpinE1 and TIMP2 correlation with RECK expression in cervical cancer. Spearman’s correlation analyses between RECK and TIMP2 or RECK and SERPINE1/PAI1 mRNA levels in both cervical cancer and HNSCC patients from TCGA database. (**C**,**D**). GO enrichment analysis focused on biological processes most associated with RECK expression in cervical cancer (**C**) and HPV positive HNSCC clinical samples (**D**). Red arrows show biological processes enriched in cervical cancer and HNSCC. Orange arrow indicates the enrichment of the cell adhesion process in agreement with animal model findings.

**Figure 7 cancers-13-02217-f007:**
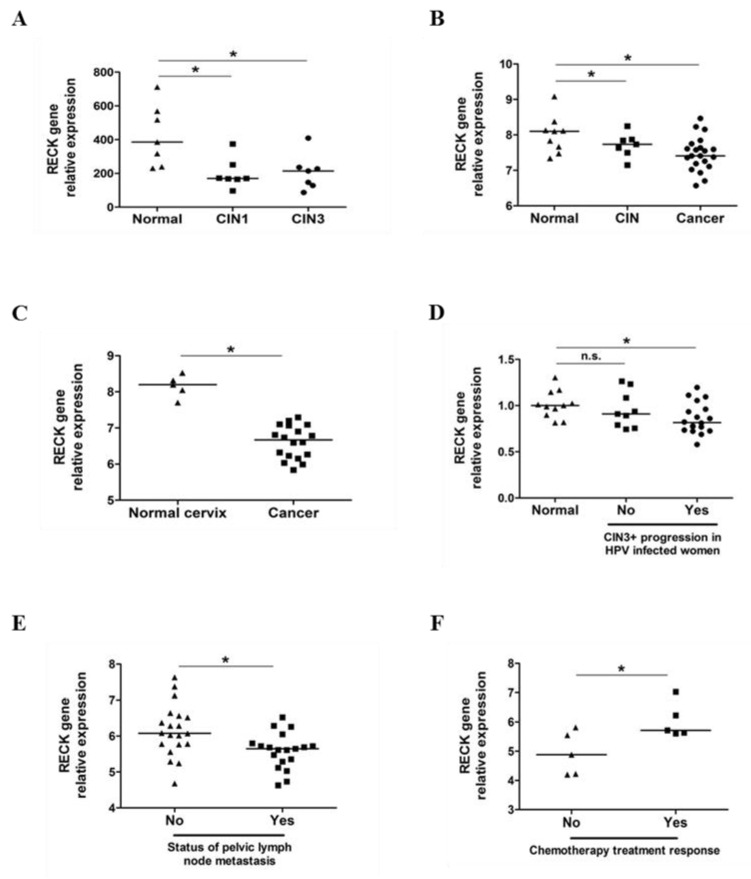
RECK expression is down regulated in cervical intraepithelial neoplasias (CIN) and invasive carcinomas being associated with worse clinical outcome of cervical cancer patients. (**A**). RECK gene expression is down regulated in CIN1 and CIN3 when compared with normal cervical tissues (GSE51993). (**B**,**C**). RECK gene expression is decreased in CIN and invasive carcinomas, when compared to normal cervical tissues (GSE7803 and GSE39001). (**D**). RECK relative expression is down regulated in women with HPV positive CIN that progressed to CIN3 or worse, when compared to women with no HPV infection (GSE75132). (**E**) RECK expression is down regulated in early-stage cervical carcinoma patients who presented pelvic lymph node metastasis, when compared to the same stage cervical carcinoma patients without metastasis (GSE26511). (**F**). RECK expression is up regulated in cervical cancer patients who presented better response to chemotherapy when compared to patients with no treatment response (GSE70035). * *p* < 0.05 by Kruskal–Wallis (7A, 7B and 7D) or Mann–Whitney (7C, 7E and 7F) test.

**Figure 8 cancers-13-02217-f008:**
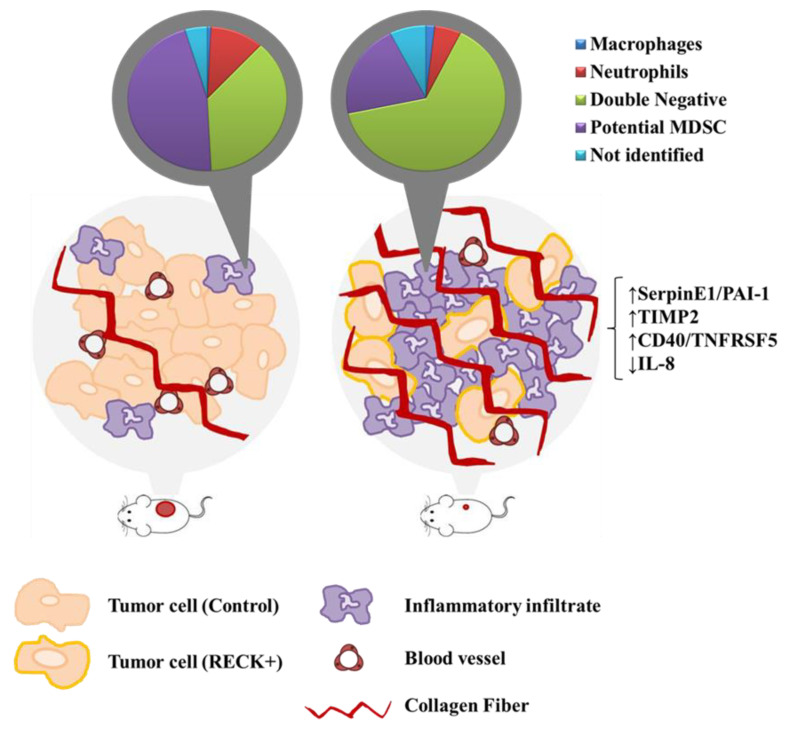
Proposed global effects of RECK expression in a cervical cancer model. Animals *s.c.* injected with cervical cancer derived cells transduced to over express RECK present delayed establishment and reduced tumor volume when compared to control. RECK over expressing cells are represented with light orange cell membrane and the red circles in mice lateral side represent the induced tumors. RECK+ tumors also showed reduced proportion of tumor and endothelial cells, the latter was depicted here in dark red blood vessels. Moreover, we observed specific increased proportion of Double negative inflammatory infiltrate (pie charts in the top) and higher collagen content (bright red linear structures) in RECK+ tumors. Finally, RECK over expression correlated with higher levels of SerpinE1/PAI-1, TIMP-2 and CD40/TNFRSF5 proteins, while presented an inverse correlation with IL-8 protein expression (on the far-right side).

## Data Availability

The datasets used and/or analyzed during the current study are available from the corresponding author on reasonable request. Links to publicly archived datasets analyzed during the study: GSE51993: https://www.ncbi.nlm.nih.gov/geo/query/acc.cgi?acc=GSE51993 (accessed on 14 April 2021); GSE7803: https://www.ncbi.nlm.nih.gov/geo/query/acc.cgi?acc=GSE7803 (accessed on 14 April 2021); GSE39001: https://www.ncbi.nlm.nih.gov/geo/query/acc.cgi?acc=GSE39001 (accessed on 14 April 2021); GSE75132: https://www.ncbi.nlm.nih.gov/geo/query/acc.cgi?acc=GSE75132 (accessed on 14 April 2021); GSE26511: https://www.ncbi.nlm.nih.gov/geo/query/acc.cgi?acc=GSE26511 (accessed on 14 April 2021); GSE70035: https://www.ncbi.nlm.nih.gov/geo/query/acc.cgi?acc=GSE70035 (accessed on 14 April 2021); Cervical Squamous Cell Carcinoma and Endocervical Adenocarcinoma (TCGA, Firehose Legacy): http://www.cbioportal.org/study/summary?id=cesc_tcga (accessed on 14 April 2021); Head and Neck Squamous Cell Carcinoma (TCGA, Firehose Legacy): http://www.cbioportal.org/study/summary?id=hnsc_tcga (accessed on 14 April 2021).

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
