# Peer review of "Low RECK Expression Is Part of the Cervical Carcinogenesis Mechanisms"

_cancers, 2021, doi:10.3390/cancers13092217_

Round 1
Reviewer 1 Report
Herbster et al. present a research focused on the role of RECK in the tumorigenesis of cervical cancer. The topic is of great interest and the work is built on sound scientific data, results are clearly presented and well connected to previous work from the same group.
The following are minor issues:
- lines 97 and 99: cells are defined "super expressing", it would be better to define them "over expressing" as in other parts of the paper
- chapters 3.2, 3.3 and 3.4 describe experiments where SiHa and SW cells sometimes have different behaviour. Authors may want to discuss this point
- Figure 4A: the percentage of the different inflammatory cells populations are quite different in the two cell lines (control). Authors may want to discuss this point
- Figure 7A: the subtypes CIN1 and CIN3 are mentioned for the first time, but neither here nor subsequently is explained what they are
- there are little typos all over the manuscripts, i.e. line 358 "my-eloid", line 365 "mono-cyte"
Author Response
We wish to deeply thank the Reviewer #1 for the careful and constructive revision of the manuscript. Below there is a point-by-point answer to issues raised by the Reviewer. Alterations in the main text are shown in red.
Response to Reviewer #1
“- lines 97 and 99: cells are defined "super expressing", it would be better to define them "over expressing" as in other parts of the paper”
The changes suggested were made.
“- chapters 3.2, 3.3 and 3.4 describe experiments where SiHa and SW cells sometimes have different behaviour. Authors may want to discuss this point.”
We thank Reviewer #1 for raising this interesting point, giving us the possibility to discuss it properly. In fact, our results show that SiHa and SW756 cells dot not exhibit identical behavior upon RECK over expression. We have not a clear explanation for this observation. However, there are several non-mutually exclusive possibilities to explain this fact. First, these cells lines were established from women with different genetic backgrounds. Second, these cells were transformed with different high-risk HPV types. Third, these cell lines may have accumulated different alterations during their time in culture. Probably, there other characteristics that may distinguish these cells. Interestingly, previous observations from our group and others have shown that different cervical cancer cell lines may exhibit different behaviors. For instance, it was observed that the kinase requirements for survival were more than similar between CasKi (HPV16) and HeLa (HPV18) cells, than between any of these cells and SiHa (HPV16) (https://doi.org/10.1073/pnas.0808019105). Besides, we observed that the expression pattern of genes involved in DNA damage response was more similar between SiHa (HPV16) and C33A (HPV-negative) cervical cancer cells than between any of these cells and HeLa (HPV18) (https://dx.doi.org/10.1038%2Fs41598-018-37064-x). From these results we can conclude that differences in the behavior of cervical cancer cell lines are not unexpected.
Grueneberg DA, Degot S, Pearlberg J, Li W, Davies JE, Baldwin A, Endege W, Doench J, Sawyer J, Hu Y, Boyce F, Xian J, Munger K, Harlow E. Kinase requirements in human cells: I. Comparing kinase requirements across various cell types. Proc Natl Acad Sci U S A. 2008 Oct 28;105(43):16472-7. doi: 10.1073/pnas.0808019105. Epub 2008 Oct 23. PMID: 18948591; PMCID: PMC2575444.
Prati B, da Silva Abjaude W, Termini L, Morale M, Herbster S, Longatto-Filho A, Nunes RAL, Córdoba Camacho LC, Rabelo-Santos SH, Zeferino LC, Aguayo F, Boccardo E. Three Prime Repair Exonuclease 1 (TREX1) expression correlates with cervical cancer cells growth in vitro and disease progression in vivo. Sci Rep. 2019 Jan 23;9(1):351. doi: 10.1038/s41598-018-37064-x. PMID: 30674977; PMCID: PMC6344518.
“- Figure 4A: the percentage of the different inflammatory cells populations are quite different in the two cell lines (control). Authors may want to discuss this point.”
That is correct, there are some differences in the inflammatory infiltrates among SiHa and SW756 tumors. The frequency of neutrophils is similar, but while in SiHa tumors we found macrophages (Ly6C+F4/80+) as the main infiltrating population, in SW756 we found that the major infiltrating population was Ly6C+Ly6G+. We did not investigate the mechanisms behind this observation, but we can make some considerations:
- a) data in the literature shows that the roles of myeloid populations in the tumor microenvironment may be, at least in part, redundant; so that these differences may not be functionally relevant. For example, work published by Pahler and collaborators (2008) showed that in CCR2 (chemokine receptor with a key role in monocyte recruitment to tumors) deficient mice, HPV associated tumors were infiltrated by neutrophils, instead of macrophages, and displayed the same functions as macrophages regarding angiogenesis;
- b) work published by Van Ginderachter and collaborators (2006) showed that CD11b+Gr1int (Gr1 antibody recognizes both Ly6C and LyG) myeloid cells that accumulated in the spleens of tumor bearing mice were the precursors of tumor associated macrophages. Therefore, it is possible that our observation may simply reflect myeloid populations that reach different levels of differentiation in the tumor microenvironment;
- c) in line with the last observation, we have shown that both SiHa and HeLa (as SW756, HeLa cells are transformed with HPV18) tumors promote myeloid cell accumulation in the spleen of tumor bearing mice. However, SiHa tumors displayed a stronger effect than HeLa or C33A tumors, promoting accumulation of both CD11+Gr1hi and Gr1int cells (Stone et al, 2014). This observation can suggest that there are differences between HPV16 and HPV18 associated tumor cells in the modulation of myeloid populations, which could lead to the results observed in this manuscript.
However, given the plasticity of these cells and the data described here. We decided to only mention in the manuscript text that the difference in the myeloid populations infiltrating control tumors is probably due to different effect these cells have on myeloid populations, but that we do not believe there are important functional differences between tumor associated macrophages in SiHa tumors and potential MDSC in SW756 tumors.
The following statement was included in the main text (lines 370-373):” Although the major myeloid population infiltrating SiHa tumors were macrophages and SW756 tumors were MDSC, we believe their role may be redundant in the tumor microenvironment [31]. Of notice, however, was that…”.
Pahler JC, Tazzyman S, Erez N, Chen YY, Murdoch C, Nozawa H, Lewis CE, Hanahan D. Plasticity in tumor-promoting inflammation: impairment of macrophage recruitment evokes a compensatory neutrophil response. Neoplasia. 2008 Apr;10(4):329-40. doi: 10.1593/neo.07871. PMID: 18392134 PMCID: PMC2288539.
Van Ginderachter JA, Meerschaut S, Liu Y, Brys L, De Groeve K, Ghassabeh GH, Raes G, De Baetselier P. Peroxisome proliferator-activated receptor γ (PPARγ) ligands reverse CTL suppression by alternatively activated (M2) macrophages in cancer. Blood. 2006. 108 (2): 525–535. doi.org/10.1182/blood-2005-09-3777. PMID: 16527895.
Stone SC, Rossetti RA, Lima AM, Lepique AP. HPV associated tumor cells control tumor microenvironment and leukocytosis in experimental models. Immun Inflamm Dis. 2014 Aug;2(2):63-75. doi: 10.1002/iid3.21. PMID: 25400927 PMCID: PMC4217549.
“- Figure 7A: the subtypes CIN1 and CIN3 are mentioned for the first time, but neither here nor subsequently is explained what they are.”
We have thank reviewer #1 for this observation. The following statement was included in the main text (lines 478-480): “High-grade CIN (CIN2/3) are alterations in the cervical epithelium and are considered the precursor lesions of cervical cancer.”
“- there are little typos all over the manuscripts, i.e. line 358 "my-eloid", line 365 "mono-cyte"”
The text has been revised thoroughly to correct errors.

Reviewer 2 Report
The study showed the inflammatory infiltrate. We would like to see if any HPV E6 or E7 antigen-specific immune responses. Any immunotherapy and targeted therapies combinations are developing to flight the cervical cancers?
Author Response
We wish to deeply thank the Reviewer #2 for the careful and constructive revision of the manuscript. Below there is a point-by-point answer to issues raised by the Reviewer. Alterations in the main text are shown in red.
Response to Reviewer #2
“The study showed the inflammatory infiltrate. We would like to see if any HPV E6 or E7 antigen-specific immune responses. Any immunotherapy and targeted therapies combinations are developing to flight the cervical cancers?”
We thank Reviewer #2 for the opportunity of discussing this interesting point. The model that we used, based on immunodeficient animals, does not allow us to test the hypothesis of an antigenic-specific response. At present, we are working on an immunocompetent mouse model that will probably allow us to address this issue properly.
There are several studies investigating immunotherapy and targeted therapies combinations for cervical cancer patients. Several strategies are under investigation: T cell adoptive transfer, therapeutic vaccines, antibodies against immune checkpoint blockade molecules, Her2 inhibitor, PARP inhibitor and others (Mauricio et al 2021). However, none have been incorporated in the state of art treatment for cervical cancer. This means that there is necessity and opportunity for the development of combination strategies or even development of new therapy approaches.
Mauricio D, Zeybek B, Tymon-Rosario J, Harold J, Santin AD. Immunotherapy in Cervical Cancer.
Curr Oncol Rep. 2021 Apr 14;23(6):61. doi: 10.1007/s11912-021-01052-8. PMID: 33852056

Reviewer 3 Report
In this manuscript, Herbster et al. follow up on previous work showing that RECK is a potential tumor suppressor in cervical cancer cells. They show using mouse models that RECK inhibits the invasive properties of cervical cancer cells in vivo. This is a well performed study and I only have a few minor comments.
Minor
- The introduction is lack in many key reports on MMP expression in cervical cancer and how HPV regulates MMPs. For example, E6 is know to regulate MMP expression via IL-8 (Shiua et al., Plos One 2013) and JNK/c-Jun (Morgan et al., Cell Death Diff., 2020) and E7 up-regulates AKT-mediated MMP expression (Basukala et al., Plos Pathogens, 2019). Furthermore, Rac signalling Such studies should be included to put this study in the wide context of the known literature.
- The data suggest that the effect of RECK re-introduction has a greater effect on tumour growth in HPV16+ SiHa cells than HPV18 SW756 cells, particularly in figure 1. The authors should look in to see if this is true across several HPV16+ and HPV18+ cells lines.
- Figure 7 - the data presented are not relative gene expression, they are gene expression using units specific to the platform used. The axes should be labelled 'gene expression (arbitrary units)
Minor
- line 76 - 'already' should be 'previously'
- The whole blot provided in Supp Figure 1C does not line up as in the cropped blot - hopefully, this is just an honest mistake...
- line 238 - '...down-regulation in important...' should be IS important
- line 240 - in vivo should be in vivo
- line 241 - injected, not in-jected
Author Response
We wish to deeply thank the Reviewer #3 for the careful and constructive revision of the manuscript. Below there is a point-by-point answer to issues raised by the Reviewer. Alterations in the main text are shown in red.
Response to Reviewer #3
We thank Reviewer #3 for the critical review of the manuscript.
“The introduction is lack in many key reports on MMP expression in cervical cancer and how HPV regulates MMPs. For example, E6 is know to regulate MMP expression via IL-8 (Shiua et al., Plos One 2013) and JNK/c-Jun (Morgan et al., Cell Death Diff., 2020) and E7 up-regulates AKT-mediated MMP expression (Basukala et al., Plos Pathogens, 2019). Furthermore, Rac signalling Such studies should be included to put this study in the wide context of the known literature.”
Following Reviewer #3 advice and in order to better contextualize our study we have added the following statement in the main text (Lines 67-75) “The direct effect of HPV oncogenes on the expression of different ECM components has been previously reported. A recent study demonstrated that c-Jun inhibition in HPV-transformed cell lines (HeLa and CasKi) was associated with lower metalloproteinases type 9 (MMP-9) mRNA expression levels [8]. Furthermore, Shiau and coworkers showed that HPV16 E6 induced MMP-2 and -9 mRNA expression levels through an IL-8 dependent pathway in H1299 cells [9]. Finally, phosphorylation of HPV18 E7 at S32 and S34 amino acids was associated with the up regulation of MMP-1 and -13 protein expression in C4-1 cervical cancer cells [10]. An extensive description of the main alterations in ECM components during HPV-associated transformation can be found elsewhere [6].”
“The data suggest that the effect of RECK re-introduction has a greater effect on tumour growth in HPV16+ SiHa cells than HPV18 SW756 cells, particularly in figure 1. The authors should look in to see if this is true across several HPV16+ and HPV18+ cells lines.”
This is a very interesting observation. In fact, we re-introduced RECK in HPV18+ HeLa cells. However, we observed that these cells proliferation was greatly affected by RECK over overexpression, precluding us from performing additional experiments with this cell line. This observation indicates that HeLa cells, although transformed by HPV18, are highly sensitive to RECK re-introduction. This suggests that RECK effect is, at least, partially independent of the HPV type present.
We also re-introduced RECK in HPV-negative cervical cancer derived cell line C33A and observed that this protein overexpression also affected the tumorigenic potential of these cells both in vitro and in vivo.
“Figure 7 - the data presented are not relative gene expression, they are gene expression using units specific to the platform used. The axes should be labelled 'gene expression (arbitrary units)”
We really thank Reviewer #3 for detecting this important mistake. The labelling of the axes has been changed as suggested.
“line 76 - 'already' should be 'previously'”
The text has been changed.
“The whole blot provided in Supp Figure 1C does not line up as in the cropped blot - hopefully, this is just an honest mistake...”
Thank you to Reviewer #3 for raising this concern and give us the opportunity to make clear how the figure was mounted. In the present version of the manuscript, we have included a detailed description of how Supplementary figure 1C was mounted. It is important to note that the PVDF membrane was cut between 75KDa and 50 KDa markers before incubation with primary antibodies. Finally, we slightly increased the contrast of blots in Supplementary figure 1C to show that the image was mounted using non-contiguous blots.
“line 238 - '...down-regulation in important...' should be IS important”
The change was made (now in line 248-249, due to de addition of the text in lines 67-75)
“line 240 - in vivo should be in vivo”
Change made (line 250)
“line 241 - injected, not in-jected”
Change made (line 253)
